# Prevalence of Type 2 Diabetes and Its Association with Added Sugar Intake in Citizens and Refugees Aged 40 or Older in the Gaza Strip, Palestine

**DOI:** 10.3390/ijerph17228594

**Published:** 2020-11-19

**Authors:** Majed Jebril, Xin Liu, Zumin Shi, Mohsen Mazidi, Akram Altaher, Youfa Wang

**Affiliations:** 1Global Health Institute, School of Public Health, Xi’an Jiaotong University Health Science Center, 76 West Yanta Road, Xi’an 710061, China; majed_jebril@outlook.com; 2Human Nutrition Department, College of Health Sciences, QU Health, Qatar University, Doha 2713, Qatar; zumin@qu.edu.qa; 3Department of Twin Research and Genetic Epidemiology, King’s College London, St Thomas’ Hospital, Strand, London SE1 7EH, UK; mohsen.mazidi@kcl.ac.uk; 4Department of Medical Sciences, University College of Science & Technology, Khan Younis 950, Palestine; a.taher@ucst.edu.ps

**Keywords:** Gaza Strip, Palestinian, added sugar intake, type 2 diabetes

## Abstract

Background: Little is known about the prevalence and risk factors of diabetes among Gaza Palestinians, 64% of whom are refugees with exceeded sugar intake. We aimed to estimate the prevalence of type 2 diabetes (T2D) and its association with added sugar intake among residents, with regular visits to primary healthcare centers (PHCs) across Gaza. Methods: From October to December of 2019, a cross-sectional survey was conducted among 1000 citizens and refugees in nine PHCs selected from the five governorates of the Gaza Strip. Information on dietary intake, medical history, and other risk factors was collected by trained health workers, using structured questionnaires. Anthropometry and biochemical data were extracted from the PHC medical record system. Results: Overall, the prevalence of diagnosed T2D and undiagnosed T2D were 45.2% and 16.8%, respectively, in adults aged 42 to 74 years, with the differences among citizens and refugees (diagnosed: 46.2% vs. 43.8%; undiagnosed: 15.7% vs. 18.2%). The uncontrolled glycaemic rate was 41.9% and 36.8% for diagnosed patients in citizens and refugees, respectively. Among those without a clinical diagnosis of T2D, after multivariable adjustment, daily added sugar intake was positively associated with fasting glucose and the risk of undiagnosed T2D (odds ratio, 95% CI, highest vs. lowest intake, was 2.71 (1.12–6.54) (*p*
_for trend_ < 0.001). In stratified analysis, the associations between added sugar intake and the risk of undiagnosed T2D tend to be stronger among refugees or those with higher body mass index. Conclusions: Among Palestinian adults, both citizens and refugees are affected by T2D. Added sugar intake is associated with the risk of undiagnosed T2D.

## 1. Introduction

Type 2 diabetes (T2D) and its complications have contributed to the global burden of mortality [1], posing a major global health threat in both developed and developing countries [2,3]. In the Middle East, approximately 46 million individuals were suffering from T2D from 2000 to 2018 [4]. There is a lack of information about the prevalence of T2D in Palestine [5], possibly due to the large population of scattered refugees across neighbouring countries, the presence of conflicts, wars, ongoing crises, and the fragile nature of the healthcare system [6].

According to the 2018 annual health report of the United Nations Relief and Works Agency (UNRWA), 15.7% of refugees aged over 40 years were diagnosed with T2D in the Gaza Strip, Palestine [7]. Moreover, T2D and its complications account for approximately 5.7% of all Palestine deaths [8]. In the past 20 years, people in the Gaza Strip have been facing poor surroundings and critical situations due to sieges and border-closure conditions [9,10]. Refugees account for 64% of all Palestinian residents in the Gaza Strip [11] and suffer from deteriorating living conditions, high poverty, low wages, and unhealthy camp housing [12], with raised problems in malnutrition, food insecurity, and lack medical care [13,14]. However, direct population evidence linking such exposures with the prevalence of T2D was rarely reported [15].

Suboptimal diet is one of the leading causes of poor health, including T2D [16,17]. In Gaza, most Palestinian refugees receive monthly food supplies (sugar, rice, legumes, flour, and cooking oil) and aids from humanitarian agencies, such as UNRWA and World Food Programme [18]. Moreover, Palestinians consume large amounts of homemade desserts, cakes, sweetened tea, and coffee [19]. A community-based cross-sectional survey undertaken among Palestinians aged 30–65 years in the West Bank showed that the mean annual intake of added sugar per household member was 37.8 kg (≈105 g/day), contributing to 13–14% daily total energy intake [20]. It is approximately three times what the American Heart Association recommends for the maximum amount of daily added sugar intake (37.5 g for men and 25 g for women) [21].

Moreover, World Health Organization (WHO) suggests reducing free sugar consumption to below 10% of total energy intake, at all stages of life, to minimize the risk of excessive weight gain [22], and further reduction to 5% could have additional health benefits [23]. On the other hand, diabetic patients with high added sugar intake can develop diabetic foot ulcers in the long-term, especially for elderly adults [24]. It may have a negative impact on the quality of life and diabetes self-management [25]. Thus, the diabetic patient should have special management of his/her nutrition, including added sugar, which has a critical role in the wound-healing process [26].

To the best of our knowledge, the link between sugar intake and the prevalence of T2D among Palestinians has not been explored. We aimed to evaluate the prevalence of T2D in Palestinian citizens and refugees living in Gaza who attended the clinics and further explored the association of added sugar intake with the risk of T2D.

## 2. Materials and Methods

### 2.1. Study Design and Participants

This is a cross-sectional study among residents who were regularly visiting the primary healthcare centers (PHCs), the Ministry of Health (MOH), Gaza Strip, Palestine. The project was designed to examine the prevalence of non-communicable diseases and conventional population-specific (e.g., living conditions under long-lasting border conflicts) risk factors among Palestinian adults in the Gaza Strip and served by the governmental PHCs.

The inclusion criteria included the following: (1) Palestinian residents (refugees or citizens), (2) date of birth between 1 January 1944 and 31 December 2001, (3) living in the Gaza Strip for 14 years; and (4) registered at PHCs, Gaza. Exclusion criteria were as follows: (1) pregnant women and (2) lacking data of existing biochemical variables from 2013 to 2019. Setting a precision of 0.05, assuming a prevalence of T2D of 15.7% [7,27], we concluded that 956 participants would be sufficient to provide 85% power to estimate the prevalence of T2D. Out of the total, 1000 participants were finally included in the analysis, after excluding pregnant participants (*n* = 37) and participants whose biochemical and anthropometric data were missing (*n* = 83). The participants of the research were chosen from various clinics within the PHCs. We invited all the visitors during the working hours of those clinics, from the early morning until afternoon.

All subjects gave their informed consent for inclusion before they participated in the study. The study was approved by the Ethical Committee of Xi’an Jiaotong University Health Science Center, and it was conducted by following the Declaration of Helsinki. The protocol was approved by the Ethics Committee of the Research Department at the Directorate General of Human Resources Development, Ministry of Health, Gaza (PHRC/HC/576/19).

### 2.2. Sampling Approach

The Gaza Strip is divided into five governorates: North of Gaza, Gaza City, Midzone, Khan-Younis, and Rafah [28]. A total of 51 PHCs are distributed across the Gaza Strip, which provides residents (both citizens and refugees) specialized healthcare services, including tailored maternal and family-planning programs, as well as permanent monitoring of non-communicable diseases (NCDs) such as T2D [27]. We selected 9 out of 51 PHCs, the largest ones located in different governorates across the Gaza Strip, as our study sites, to maximize our participants’ representativeness. One to three PHCs were consequently selected from each governorate, matching the population’s size of each governorate (Appendix A
Appendix A).

### 2.3. Data Collection

The initial version of the questionnaire was designed and then validated in two steps. First, the research instrument was sent to medical experts, to express its simplicity, relativity, and importance. Secondly, a pilot study was conducted among a small group of PHCs’ visitors (*n* = 30), to verify the validity and reliability of the data-collection process before the formal data collection. Minor changes were made to the data-collection approach for some items, by considering the results of the pilot analysis. Furthermore, medical experts validated and reviewed the updated questionnaire, to reconfirm the acceptability of the methods used to achieve the research goals. In the formal data-collection process, the final version of the questionnaire was used, and the data from the pilot study were not used in the final analysis.

The data were collected in two stages: (1) Face-to-face questionnaire interviews were conducted for all participants, by trained health workers. (2) Anthropometric and biochemical parameters for the same respondents were extracted by medical secretary staff, using their Electronic Health Record System (E-Health), by matching their registration ID.

The questionnaire consisted of four parts: (1) sociodemographic data, including age, sex, marital status, education level, working status, and household income; (2) medical history, including diagnosis, duration, treatment status, and family history of non-communicable disease, including T2D; (3) lifestyle information, including physical activity (International Physical Activity Questionnaire [29]), dietary habits (adapted from questions used in the China Kadoorie Biobank [30]), tea and coffee consumption, and smoking; (4) unique issues for the poor conditions of the Palestinian refugees and citizens, such as living, housing conditions, injuries, loss of family members, and depression.

The Anthropometric variables, including weight, height, waist circumference, and blood pressure, were extracted from the E-Health record system. Body mass index (BMI) was calculated based on the BMI formula (BMI (kg/m^2^) = Weight (kg)/Height squared (m^2^)) [31]. In the daily routine work of PHCs, the nursing staff measured height and weight for all clinic visitors, using stadiometers (Seca), according to the WHO standardized protocols [31]. Waist circumference was routinely measured to the nearest 1.0 cm, after breath out, using non-elastic tape around the middle waist, just above the hipbones, according to the protocol of the National Institutes of Health [32]. Blood-pressure data were extracted from the E-Health system. Their blood pressure was measured by the PHCs nursing staff, in the right arm, after 5 min of rest, in a sitting position, by a single mercury sphygmomanometer.

Biochemical data were extracted from the E-health system: fasting blood glucose (FBG), Haemoglobin A_1c_ (HbA_1c_), total cholesterol, triglycerides, high-density lipoprotein (HDL), and low-density lipoprotein (LDL). All of these biochemical tests were assayed by medical laboratory technicians, using colorimetric and kinetic assays, using an automated biochemical analyzer (ChemWell 2910, Awareness Technology, CA, USA and Erba XL 200, Erba Diagnostics Mannheim, Germany).

### 2.4. Outcome Definitions

T2D was defined as having a documented diagnosis by general practitioners in the PHCs, and those cases were named diagnosed T2D in the present study. Among those without a clinical T2D diagnosis, undiagnosed T2D was defined as having a fasting blood glucose (FBG) level ≥7.0 mmol/L (≥126 mg/dL). Moreover, in a sensitivity analysis, for those without a clinical diagnosis of T2D, FBG was ≥7.0 mmol/L (≥126 mg/dL) plus HbA_1c_ ≥48 mmol/mol (≥6.5%). Uncontrolled diabetes was defined as having HbA_1c_ ≥53 mmol/mol (≥7.0%), according to the American Diabetes Association [33].

### 2.5. Statistical Analysis

All analyses were conducted by using STATA 14.0 (StataCorp, College Station, TX, USA). Continuous variables were reported as mean values and standard deviations, while categorical variables were presented as frequency and percentage. Chi-square and t-tests were used for comparison of categorical data and continuous variables, respectively. To compare demographic characteristics between groups, one-way variance analysis (ANOVA) was used if there were three or more groups. The odds ratio (OR) and 95% confidence interval (CI) of T2D were calculated by using multivariable logistic regression by average daily added sugar intake, after adjusting for sex, age, population (refugees, citizens), region (North, Gaza City, Midzone, Khan Younis, and Rafah), household income (<500 NIS, 500–1000 NIS, 1000–1500 NIS, >1500 NIS, and no constant income), education (low, moderate, or high), smoking (yes or no), BMI (continuous), physical activity (low, moderate, or high), fruits and vegetable intake (continuous), hypertension (having or not), and lipid profile (total cholesterol, triglycerides, HDL, and LDL). Stratified analysis was conducted by sex, age (according to the median ≥ or < 59 years), BMI (according to the median ≥ or < 26.7 kg/m^2^), and population type (refugees or citizens); the statistical significance level was set as a two-sided *p* < 0.05.

## 3. Results

### 3.1. Prevalence of T2D and Characteristics of the Participants

The mean age of the participants was 59.2 ± 7.5 years, and 46.8% were females. In total, 42.2% of the participants were refugees, and 45.1% had a low education level. Furthermore, 85.4% of them lacked constant income, and only 11.7% were employed (Table 1).

Overall, 45.2% of the participants were clinically diagnosed with T2D, and another 16.8% were undiagnosed T2D. (Table 1) Moreover, clinically diagnosed T2D tended to be more prevalent in women, age (40–50 years), citizens, unemployed, or from the north of Gaza, relative to their counterparties, while undiagnosed T2D cases were more prevalent in women, age (40–50 years), refugees, Gaza City, or employed. (Figure 1) Furthermore, 39.8% (95% CI 35.4–44.4) of the diagnosed T2D cases were uncontrolled; 40.8% (95% CI 34.7–47.3) and 38.7% (95% CI 32.4–45.4) in men and women (Figure 2); the highest uncontrolled glycaemic rate was 50.0% (95% CI 36.5–63.4) in Gaza City, while the lowest was 32.3% (95% CI 23.7–42.3) in Khan Younis. Specifically, 11.3% (95% CI 9.5–13.4) had HbA_1c_ value of 53 mmol/mol (7.0%) to 63 mmol/mol (7.9%), while the highest percentage was 17.3% (95% CI 15.1–19.7) for those who had HbA_1c_ value of 64 mmol/mol (8.0%) to 75 mmol/mol (9.0%), and 11.1% (95% CI 9.3–13.2) had HbA_1c_ value higher than 75 mmol/mol (>9.0%). (Figure 2).

Compared with non-cases, participants diagnosed or undiagnosed with T2D had higher levels in adiposity index, FBG, and HbA_1c_, and worse lipid profiles, where the mean (SD) BMI of the diagnosed and undiagnosed T2D was 28.2 (3.7) kg/m^2^ and 28.5 (3.5) kg/m^2^, respectively, while it was 27.9 (3.7) kg/m^2^ among the non-cases group (*p* = 0.189, Table 1), also the mean (SD) FBG of the diagnosed and undiagnosed T2D was 12.5 (4.6) mmol/L, and 10.0 (3.1) mmol/L, respectively, while it was 5.6 (1.0) mmol/L among the non-cases group (*p* < 0.001, Table 1).

No statistical differences were detected for population, region, education, income, working status, smoking, and blood pressure values among groups, except for sex (*p* = 0.034, Table 1) and physical activity (*p* = 0.044, Table 1). Moreover, no significant differences were observed among refugee and citizen groups (Appendix A
Appendix A).

### 3.2. Associations between Added Sugar Intake Level and Glycaemic Parameters

The mean (SD) of added sugar intake was 60.6 (22.4) g/day, and 44.1% had added sugar intake above 50 g/day. In all participants, no association was observed between added sugar intake and diagnosed T2D (*p* = 0.338, Appendix A
Appendix A). We evaluated the associations of added sugar intake and glycaemic indices (FBG and HbA_1c_) only among those without a clinical diagnosis of T2D. After adjustment of age, sex, population, region, income, education, smoking, and physical activity, higher daily added sugar intake was significantly associated with higher levels of FBG and HbA_1c_ (*p* = 0.007 for FBG, and *p* < 0.001 for HbA_1c_, Figure 3). The same pattern was observed after including physical activity and BMI in the model, only for HbA_1c_ (*p* = 0.036) but not FBG (*p* = 0.067, Figure 3).

### 3.3. Associations between Added Sugar Intake and the Risk of Undiagnosed T2D

With multiple adjustments, including age, sex, population, region, income, education, and smoking, compared with the lowest added sugar intake (<50.0 g/day), the ORs (95% CI) of undiagnosed T2D were 2.13 (1.22–3.72) for the 50.0–66.0 g/day group, 1.86 (1.06–3.25) for the 67.0–99.0 g/day group, and 3.52 (1.56–7.93) for the ≥100.0 g/day group (*p* for trend < 0.001). The association strength was not substantially changed after further adjusting for physical activity and BMI in Model 2, fruits and vegetable intake in Model 3, and hypertension plus lipid profile in Model 4 (Table 2). A sensitivity analysis was conducted among those without a clinical diagnosis of T2D. We defined undiagnosed T2D based on both FBG and HbA_1c_. The overall results were similar (*p* for trend < 0.001, Appendix A
Appendix A).

In the stratified analysis by BMI (Table 3), the added sugar intake level was positively associated with risks of undiagnosed T2D (*p* for interaction = 0.001, Table 3) only in those with a BMI ≥ 26.7 kg/m^2^, where ORs were 6.83 (2.69–17.36) for the 50.0–66.0 g/day group, 4.72 (1.91–11.64) for the ≥67.0 g/day group (*p* for trend = 0.037), but not in those with BMI < 26.7 kg/m^2^. Interestingly, when stratified by population type (refugees or citizens), the added sugar intake level was positively associated with risks of undiagnosed T2D (*p* for interaction = 0.047, Table 3) only in refugees, where ORs were 5.46 (1.75–17.08) for the 50.0–66.0 g/day group, 5.09 (1.64–15.84) for the ≥67.0 g/day group (*p* for trend = 0.043), but not in citizens. Meanwhile, in the sensitivity analysis, we defined undiagnosed T2D based on both FBG and HbA_1c_, and the findings were similar (Appendix A
Appendix A).

## 4. Discussion

In the present study, we estimated the prevalence of T2D among Gaza Palestinian adults, and it showed that higher added sugar consumption is strongly associated with the risk of undiagnosed T2D. This association was stronger among those with a higher BMI or refugees.

The estimated prevalence of diagnosed T2D was 45.2%, higher than a report on the prevalence of T2D by UNRWA (15.7%) [7], which serves only refugees, while our study included both refugees and citizens. Overall, T2D is a considerable burden on the Gazans, in light of the daily living crises that all Gazans live in [34]. Such crises include high poverty, low wages, and unhealthy camp housing [35] experienced by the Gaza residents, especially refugees [12,36]. Furthermore, the long durable siege in the Gaza Strip since 2006 [37], in addition to the three wars in Gaza (2008/2009, 2012, and 2014) [38], have significantly impacted the health sector and the quality healthcare services [39]. Those traumatic life events might make the Palestinians experience prolonged stress-related disorders [40], leading to neuroendocrine dysfunction and increasing the risk of disruption in pancreatic β cells, insulin sensitivity, and insulin secretion [41], or chronic low-grade inflammation [42], contributing to the development of T2D [43,44,45]. On the other hand, it is already mentioned in the literature that aging influences the health-related quality of life for diabetic patients [46]. A survey also suggested that T2D patients aged 65 to 84 years had a poorer health-related quality of life (HRQOL) than the general population, especially regarding different aspects of physical functioning and emotional disturbances [47]. Additionally, older adults with diabetes have unique issues that may lead to deterioration in their self-care behaviors [48].

Our study showed that glycaemic control was suboptimal for patients diagnosed with T2D, and the highest uncontrolled glycaemic rate was 50.0% (95% CI 37–63) in Gaza City. The lowest was 32.3% (95% CI 24–42) in Khan Younis. It is reasonable that Gaza City is considered an urbanized City compared to the rest of the Gaza Strip regions, while Khan Younis is a Palestinian refugee camp [49]. Most of the residents in Gaza City are citizens, not refugees [50]. Overall, it has appeared that citizens are relatively less successful in controlling glycaemia than refugees (Figure 2). One of the most important reasons might be that Palestinian citizens are less likely to have health insurance. In contrast, Palestinian refugees receive more health services over Palestinian citizens [51,52]. Similarly, in a study among 1308 diagnosed T2D seeking care at four main PHCs in the Southern West Bank of Palestine, only 16.1% had HbA_1c_ < 53 mmol/mol (<7.0%) [5], showing a very poor glycaemic control in line with our results. In another cross-sectional study in the Gaza Strip (*n* = 369 T2D patients), only one-fifth of the patients had good glycaemic control (HbA_1c_ < 53 mmol/mol (<7.0%)) [53]. The poor glycaemic control in our study might also be related to the presence of suboptimal diets, extremely low physical activity, and the lack of income [53,54,55].

Furthermore, even at the health system in Gaza, there is a shortage of essential drugs, due to the blockade’s impact on Gaza [56]. The stock of medications in Gaza mainly depends on foreign support for securing treatment, which is not permanent [57].

We found that Palestinian adults in Gaza consumed approximately 61.0 g/day of added sugar per household member, which is above the recommendations of the American Heart Association [21]. In a community-based cross-sectional study conducted among Palestinians (aged 30–65 years) in the West Bank, the annual intake of added sugar per household member was 37.8 kg (≈105.0 g/day), and added sugar intake contributed to 13–14% daily total energy intake among Palestinians (aged 30–65 years) in the West Bank, according to the results of a community-based cross-sectional study [20]. Additionally, an ecological epidemiology study reported a strong positive correlation between per capita added sugar consumption and the prevalence of T2D, whereas Asian countries had the highest correlation coefficients [58]. Therefore, as far as we know, our study is the first study reporting a direct association between pure sugar intake and diabetes risk.

There are growing concerns about the possible health threat posed by high added sugar intake, as evidenced by the latest American Dietary Guidelines [59]. The excess of sugar intake is an emerging factor in the current epidemic of obesity and metabolic-associated diseases [60,61]. Interestingly, we observed a stronger association between added sugar intake with undiagnosed T2D among refugees. Refugees in Gaza consume relatively more sugar than the citizens (Table 3 and Appendix A
Appendix A), despite their poor living and nutritional conditions [62]. About 72% of the Palestinian refugees in the Gaza Strip receive food supply aids monthly or every three months from humanitarian agencies, mainly from the UNRWA, World Food Programme, and charity organizations to secure their food or part of their essential food [18,63]. Such food aids from just UNRWA to each member in one household include commodities of sugar, flour, cooking oil, powdered milk, and others [64], and all of these quantities are doubled by UNRWA, for the benefit of the most miserable refugee families [51]. High consumption of added sugar exceeds the energy requirement of refugees who have limited physical activity levels, which may result in increased de novo lipogenesis [64], increased free fatty acids [65,66,67], central obesity, and insulin resistance [68,69]. More studies are needed to elucidate the mechanisms.

In our stratified analysis by BMI, added sugar intake level was strongly associated with undiagnosed T2D among those without a clinical diagnosis of T2D, with higher BMI (≥26.7 kg/m^2^) rather than those with a lower BMI (<26.7 kg/m^2^). The concern here is developing to T2D among those without a clinical diagnosis of T2D, with a higher BMI and consuming more sugar than those with a normal BMI. Patients with a higher BMI have the chance of elevation of free fatty acids (FFAs), mainly due to the release of increased fat mass [70]. Further studies are needed to elucidate whether and how obesity may amplify the effects of excess added sugar intake on insulin action.

Our study has some limitations: First, cross-sectional data do not explore the causal pathways that underlie the reported association. Second, selection bias is possible, as we recruited only those visitors who regularly visited the PHCs. Third, recall bias is also possible by using food-frequency questionnaires. Despite the above limitations, this is the first study that focused on a unique population (exposed to exceptional circumstances of war, instability, and ongoing violence) to determine the prevalence of T2D among refugees and citizens.

## 5. Conclusions

This study involved both citizen and refugee adults aged 40 or older in Gaza Palestinians who had regularly visited the PHCs, with an estimated prevalence of 45.2% and 16.8%, respectively, for diagnosed and undiagnosed T2D. Our study indicated that 44.1% of the population had added sugar intake above 50 g/day, with an average consumption of 60.6 g/day per household member. Furthermore, our findings also showed that higher consumption of added sugar is associated with a higher risk of undiagnosed T2D, particularly in refugees or those with higher BMI. Therefore, mass-level educational awareness campaigns are needed to provide education to Palestinians about diabetes, balanced diet, and diet-related diseases. Moreover, effective strategies should also be undertaken to reduce the diabetes burden among Gaza residents, focusing on sugar intake.

## Figures and Tables

**Figure 1 ijerph-17-08594-f001:**
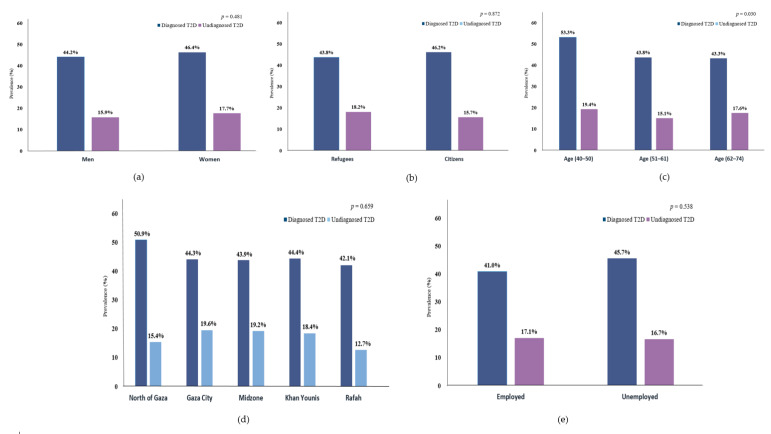
Prevalence of diagnosed and undiagnosed T2D by (**a**) sex, (**b**) population, (**c**) age groups, (**d**) region, and (**e**) working status.

**Figure 2 ijerph-17-08594-f002:**
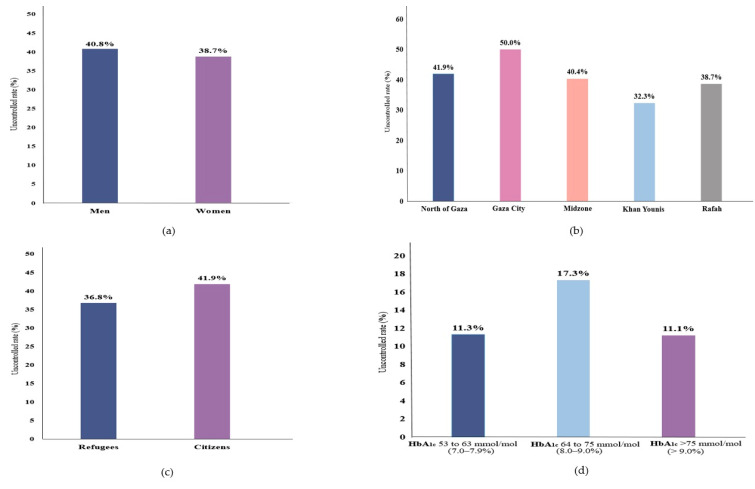
Uncontrolled glycaemic rate of T2D among diagnosed T2D by (**a**) sex, (**b**) region, (**c**) population, and (**d**) HbA_1c_ values category.

**Figure 3 ijerph-17-08594-f003:**
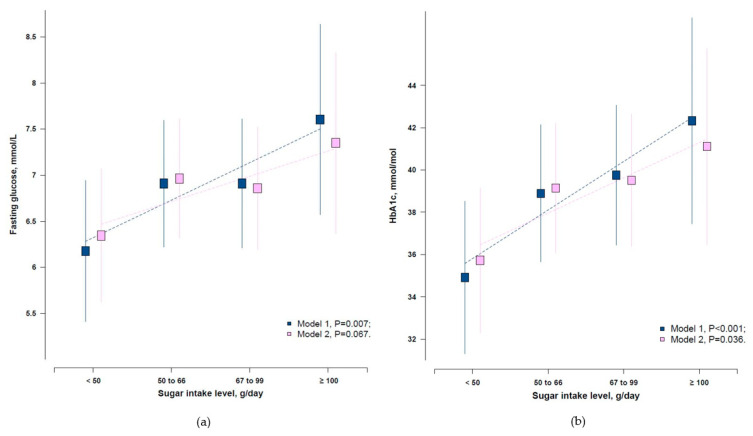
Adjusted means of (**a**) FBG and (**b**) HbA_1c_ by added sugar intake level among those without a clinical diagnosis of T2D. Model 1, Adjusted for age, sex, population, region, income, education, and smoking; Model 2, Model 1 plus physical activity and body mass index.

**Table 1 ijerph-17-08594-t001:** Characteristics of study participants.^a.^

Variable	All *n* = 1000	Diagnosed T2D *n* = 452	Undiagnosed T2D *n* = 168	Non-Cases *n* = 80	*p*-Value
Age, years	59.2 (7.5)	59.1 (7.7)	58.5 (7.3)	59.4 (7.5)	0.455
Women	468 (46.8%)	217 (46.4%)	83 (17.7%)	168 (35.9%)	0.034
Study regions North	220 (22.0%)	101 (22.3%)	32 (19.1%)	87 (22.9%)	0.286
Gaza	122 (12.2%)	60 (13.3%)	26 (15.5%)	36 (9.5%)	
Midzone	214 (21.4%)	103 (22.8%)	25 (14.8%)	86 (22.6%)	
Khan Younis	223 (22.3%)	99 (21.9%)	41 (24.4%)	83 (21.8%)	
Rafah	221 (22.1%)	89 (19.7%)	44 (26.2%)	88 (23.2%)	
Refugees	422 (42.2%)	185 (40.9%)	76 (45.2%)	161 (42.4%)	0.625
Family size	6.9 (1.5)	6.9 (1.6)	6.7 (1.5)	6.9 (1.5)	0.099
Married	844 (84.4%)	373 (82.5%)	147 (87.5%)	324 (85.3%)	0.386
Education Low	451 (45.1%)	208 (46.0%)	73 (43.4%)	170 (44.7%)	0.873
Moderate	295 (29.5%)	134 (29.7%)	47 (28.0%)	114 (30.0%)	
High	254 (25.4%)	110 (24.3%)	48 (28.6%)	96 (25.3%)	
Employed	117 (11.7%)	48 (10.6%)	20 (11.9%)	49 (12.8%)	0.594
Household income (NIS)					0.287
<500	51 (5.1%)	26 (5.7%)	11 (6.5%)	14 (3.7%)	
500–1000	41 (4.1%)	23 (5.1%)	9 (5.4%)	9 (2.4%)	
1000–1500	48 (4.8%)	24 (5.3%)	8 (4.8%)	16 (4.2%)	
>1500	6 (0.6%)	3 (0.7%)	2 (1.2%)	1 (0.3%)	
No constant income	854 (85.4%)	376 (83.2%)	138 (82.1%)	340 (89.4%)	
Having family history of NCDs ^b^	468 (46.8%)	211 (46.7%)	83 (49.4%)	174 (45.8%)	0.735
Physical activity					0.044
Low	654 (65.4%)	306 (67.7%)	102 (60.7%)	246 (64.7%)	
Moderate	247 (24.7%)	99 (21.9%)	42 (25.0%)	106 (27.9%)	
High	99 (9.9%)	47 (10.4%)	24 (14.3%)	28 (7.4%)	
T2D Medications					
Oral hypoglycaemic drugs	225 (49.7%)	225 (49.7%)	-	-	
Insulin	171 (37.8%)	171 (37.8%)	-	-	
Both	56 (12.4%)	56 (12.4%)	-	-	
Current cigarette’s smoking	446 (44.6%)	210 (46.5%)	66 (39.3%)	170 (44.7%)	0.279
BMI, kg/m^2^	28.1 (3.7)	28.2 (3.7)	28.5 (3.5)	27.9 (3.7)	0.189
Waist circumference, cm	105.0 (14.7)	105.5 (14.8)	105.2 (14.7)	104.4 (14.6)	0.547
Fasting glucose, mmol/L	9.4 (4.7)	12.5 (4.6)	10.0 (3.1)	5.6 (1.0)	<0.001
HbA_1c_, mmol/mol	48.4 (18.8)	60.5 (17.3)	50.5 (14.4)	32.9 (8.1)	<0.001
HbA_1c_, %	6.6 (1.7)	7.7 (1.6)	6.7 (1.3)	5.2 (0.7)	<0.001
Total cholesterol, mmol/L	5.7 (1.6)	6.2 (1.7)	5.6 (1.3)	5.1 (1.4)	<0.001
Triglycerides, mmol/L	2.1 (0.8)	2.3 (0.8)	2.0 (0.6)	1.8 (0.7)	<0.001
HDL, mmol/L	1.1 (0.2)	1.0 (0.2)	1.1 (0.3)	1.1 (0.2)	<0.001
LDL, mmol/L	4.4 (1.3)	4.6 (1.3)	4.2 (1.2)	4.2 (1.2)	<0.001
Systolic blood pressure, mmHg	137.4 (21.4)	137.7 (21.1)	137.2 (20.9)	137.1 (21.9)	0.916
Diastolic blood pressure, mmHg	85.6 (7.2)	85.7 (7.1)	85.5 (7.4)	85.8 (7.3)	0.935
Added sugar intake, g/day	60.6 (22.4)	60.1 (22.0)	59.3 (23.4)	61.8 (22.5)	0.421
Fruits consumption, g/day	163.2 (49.3)	161.4 (49.4)	164.8 (48.2)	164.5 (49.8)	0.591
Vegetable consumption, g/day	282.3 (59.6)	278.8 (59.9)	285.1 (61.8)	285.0 (58.4)	0.267

^a^ Data are presented as mean (SD) for continuous measures and *n* (%) for categorical measures. ^b^ NCDs, non-communicable diseases, such as stroke, cardiovascular disease, cancer, diabetes mellitus, and hypertension, for father and mother together. NIS, New Israeli Shekel; BMI, body mass index; HbA_1c_, haemoglobin A_1c_; HDL, high-density lipoprotein; LDL, low-density lipoprotein; T2D, type 2 diabetes.

**Table 2 ijerph-17-08594-t002:** Adjusted odds ratio (95% CI) of undiagnosed T2D by average daily added sugar intake (*n* = 548).

Added Sugar Intake Level g/day	No. of Participants *n* (%)	Model 1	Model 2	Model 3	Model 4
<50.0 g/day	121 (22.1%)	1.00	1.00	1.00	1.00
50.0–66.0 g/day	183 (33.4%)	2.13 (1.22–3.72)	2.01 (1.13–3.59)	2.01 (1.12–3.59)	1.98 (1.09–3.60)
67.0–99.0 g/day	199 (36.3%)	1.86 (1.06–3.25)	1.59 (0.88–2.87)	1.60 (0.89–2.88)	1.54 (0.84–2.81)
≥100.0 g/day	45 (8.2%)	3.52 (1.56–7.93)	2.72 (1.15–6.39)	2.72 (1.15–6.39)	2.71 (1.12–6.54)
*p* for trend		<0.001	<0.001	<0.001	<0.001

Model 1, adjusted for age, sex, population, region, income, education, and smoking; Model 2, Model 1 plus physical activity and body mass index; Model 3, Model 2 plus fruits and vegetable intake; Model 4, Model 3 plus hypertension and lipid profile (total cholesterol, triglycerides, HDL, and LDL).

**Table 3 ijerph-17-08594-t003:** Odds ratio (OR) (95% CI) of undiagnosed T2D by added sugar intake level and other risk factors (*n* = 548).

Variable N (%)	<50.0 g/day OR (95% CI) *n* (%)	50.0–66.0 g/day OR (95% CI) *n* (%)	≥67.0 g/day OR (95% CI) *n* (%)	*p*-Value _trend_	*p*-Value _interaction_
Sex					0.143
Men	1.00	1.54 (0.67–3.55)	2.04 (0.93–4.47)	0.058	
310 (56.6%)	60 (19.4%)	104 (33.5%)	146 (47.1%)		
Women	1.00	2.67 (1.19–6.02)	1.42 (0.63–3.18)	0.989	
238 (43.4%)	61 (25.6%)	79 (33.2%)	98 (41.2%)		
Age (Median)					0.318
<59 years	1.00	1.69 (0.71–4.03)	1.64 (0.73–3.69)	0.044	
259 (47.3 %)	59 (22.8%)	79 (30.5%)	121 (46.7%)		
≥59 years	1.00	1.41 (0.62–3.19)	1.06 (0.46–2.40)	0.679	
289 (52.7%)	62 (21.4%)	104 (36.0%)	123 (42.6%)		
BMI (Median)					0.001
<26.7 kg/m^2^	1.00	1.94 (0.75–5.00)	2.17 (0.84–5.34)	0.808	
216 (39.4%)	52 (24.1%)	70 (32.4%)	94 (43.5%)		
≥26.7 kg/m^2^	1.00	6.83 (2.69–17.36)	4.72 (1.91–11.64)	0.037	
332 (60.6%)	69 (20.7%)	113 (34.1%)	150 (45.2%)		
Population					0.047
Refugees	1.00	5.46 (1.75–17.08)	5.09 (1.64–15.84)	0.043	
237 (43.2%)	52 (21.9%)	77 (32.5%)	108 (45.6%)		
Citizens	1.00	4.93 (1.61–12.26)	3.98 (1.33–11.95)	0.666	
311 (56.8%)	69 (22.2%)	106 (34.1%)	136 (43.7%)		

Variables adjusted for age, sex, population, region, income, education, smoking, physical activity, body mass index, fruits and vegetable intake, hypertension, and lipid profile (total cholesterol, triglycerides, HDL, and LDL).

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
