# Peer review of "Prevalence of Type 2 Diabetes and Its Association with Added Sugar Intake in Citizens and Refugees Aged 40 or Older in the Gaza Strip, Palestine"

_ijerph, 2020, doi:10.3390/ijerph17228594_

Round 1

Reviewer 1 Report

In this study, the authors were aimed  to  evaluate  the  prevalence  of  T2D  in  Palestinian citizens and refugees living in Gaza who attended the clinics, further explore the associations of sugar intake with the risks of T2D. The study covers some issues that have been overlooked in other similar topics. The study was conducted with a good scientifically sound. Minor comments: the manuscript needs moderate English change and grammar correction. Authors should add some "take-home message" at the end of the conclusion section.

Author Response

Dear reviewer,

We appreciate the time and effort that you have dedicated to providing your valuable feedback on our manuscript. We are grateful for your insightful comments on our paper. We have incorporated changes to reflect the suggestions provided. (The page and line numbers are addressed according to the clean version of the manuscript). All changes are marked in yellow in the revised manuscript. We hope will reach your aspirations. Thank you very much.

Here is a point-by-point response to the reviewers’ comments and concerns:

Reviewer 1

In this study, the authors were aimed to evaluate the prevalence of T2D in Palestinian citizens and refugees living in Gaza who attended the clinics, further explore the associations of sugar intake with the risks of T2D. The study covers some issues that have been overlooked in other similar topics. The study was conducted with a good scientifically sound. Minor comments: the manuscript needs moderate English change and grammar correction. Authors should add some "take-home message" at the end of the conclusion section.

1. The manuscript needs moderate English change and grammar correction.

RESPONSE: Thank you for pointing this out. We agree with this comment. And based on your comment, we have revised and corrected the English mistakes in the whole manuscript carefully. (Please note these corrections in the highlighted yellow across the manuscript).

2. Authors should add some "take-home message" at the end of the conclusion section.

RESPONSE: Thank you for pointing this out. We agree with this very important suggestion.  Therefore, we have added a paragraph of “Therefore, mass level educational awareness campaigns are the need to provide education to Palestinians about diabetes, balanced diet, and diet-related diseases. Moreover, effective strategies should also be undertaken to reduce the diabetes burden among Gaza residents, focusing on sugar intake” at the end of the conclusion, (page 13; line 321-324).

Reviewer 2 Report

In this study, the authors examined prevalence of  T2D and risk factors of T2D among Gaza Palestinians, an under-studied population. Given the high prevalence of T2D and unfavorable environment in this population, this paper may have important public health implications, although the studied risk factors of T2D are well-known (e.g., sugar intake). Overall, the paper is very well-written, with sound methods, clear result presentation and thoughtful discussion. I have a few comments and suggestions for authors' consideration.

  1. T2D is an age related-disease. The high prevalence in this study population might be largely due to relative older age. The authors only provided mean age but not range of age. Prevalence of T2D across age groups would be interesting. The age of this population needs to be indicated in title and abstract, otherwise it would be misleading. 
  2. The other limitation of this study is that all participants came from pHC, which might be a population-representative sample. The prevalence of T2D could be over-estimated.
  3. In the introduction, Reference 21 about data in Canada is not very relevant which can be removed.
  4. It is unclear whether here sugar intake is total sugar or added sugar intake. It would be interesting to look at added sugar intake. 
  5. it is unclear why the authors did not look at other dietary factors, such as an overall dietary quality as well as some individual food intake (e,g, fruit, vegetables, red meat).

Author Response

Dear reviewer,

We appreciate the time and effort that you have dedicated to providing your valuable feedback on our manuscript. We are grateful for your insightful comments on our paper. We have incorporated changes to reflect the suggestions provided. (The page and line numbers are addressed according to the clean version of the manuscript). All changes are marked in yellow in the revised manuscript. We hope will reach your aspirations. Thank you very much.

Here is a point-by-point response to the reviewers’ comments and concerns:

Reviewer 2

In this study, the authors examined prevalence of T2D and risk factors of T2D among Gaza Palestinians, an under-studied population. Given the high prevalence of T2D and unfavorable environment in this population, this paper may have important public health implications, although the studied risk factors of T2D are well-known (e.g., sugar intake). Overall, the paper is very well-written, with sound methods, clear result presentation and thoughtful discussion. I have a few comments and suggestions for authors' consideration.

1. T2D is an age related-disease. The high prevalence in this study population might be largely due to relative older age. The authors only provided mean age but not range of age. Prevalence of T2D across age groups would be interesting. The age of this population needs to be indicated in title and abstract, otherwise it would be misleading. 

RESPONSE: Thank you very much for your suggestions. We agree with you. The age range in our study was 42 to 72 years. Accordingly, the title has been changed to “Prevalence of Type 2 Diabetes and Its Association with Added Sugar Intake in Citizens and Refugees Aged 40 or Older in the Gaza Strip, Palestine”. Also, we have added it in the abstract that “in adults aged 42 to 74 years”, (page 1; line 27). And also, we have added age groups in figure 1 (c), (page 8). Moreover, in the text, we mentioned that clinically diagnosed and undiagnosed T2D tended to be more prevalent among those aged 40 to 50 years in both results (page 4, lines 168-170) “Moreover, clinically diagnosed T2D tended to be more prevalent in women, age (40-50 years), citizens, unemployed, or from the North of Gaza, relative to their counterparties, while undiagnosed T2D were more prevalent in women, age (40-50 years), refugees, Gaza city, or employed”, and in conclusion (page 13; line 316) as well.

2. The other limitation of this study is that all participants came from PHCs, which might be a population-representative sample. The prevalence of T2D could be over-estimated.

RESPONSE: Thank you for this comment. We agree with your opinion, there is a possibility of selection bias present in our study. Therefore, we have already mentioned this at our limitations, that there is a possibility of selection bias as we recruited only those visitors who regularly visited the PHCs, (page 13; line 310-311). “Second, there is a possibility of selection bias as we recruited only those visitors who regularly visited the PHCs”.

3. In the introduction, Reference 21 about data in Canada is not very relevant which can be removed

RESPONSE: Thank you. Done, we have removed this reference from the introduction and reference list as well.

4. It is unclear whether here sugar intake is total sugar or added sugar intake. It would be interesting to look at added sugar intake. 

RESPONSE: Thank you very much for providing this valuable comment. Actually, in our research, we asked for sugar added to food and drinks, not the total sugar.  We agree with your suggestion, and on this basis, we have changed the title to ‘‘Prevalence of Type 2 Diabetes and Its Association with Added Sugar Intake in Citizens and Refugees Aged 40 or Older in the Gaza Strip, Palestine’’. Also, we replaced ‘‘sugar intake’’ with ‘‘added sugar intake’’ in all the subtractions mentioned in the manuscript as a whole.

5. It is unclear why the authors did not look at other dietary factors, such as an overall dietary quality as well as some individual food intake (e.g., fruit, vegetables, red meat).

RESPONSE: Thank you for this suggestion. Indeed, it would have been interesting to explore this aspect and investigate the other dietary factors as well, but in our research, we wanted to focus intensively on consuming added sugar, and we did not ignore other dietary intakes, such as fruits and vegetable, we already added them in table 1 within ‘‘Characteristics of study participants’’  (Page number 6), and also we added fruits & vegetable intake as potential confounders (adjustment variables with other variables throughout models) in tables (2 and 3; Page 11), also in the supplementary tables (2, 3, and 4; Pages 2-3).

Reviewer 3 Report

Dear Authors,

Glad to have an opportunity to review this manuscript, but first of all, this manuscript cannot be accepted as its current form and format. There are severe problematic areas of the manuscript and the authors were not able to deal with the essential aspects of so-called “scientific research.”

This study cannot be progressed into any further steps of publication in this quality journal unless the following issues properly dealt with:

The abstract not successfully compiling and summarizing focal points of this study

The authors were not successfully provide adequate rationales for their comments in many of sentences especially when they commented about the selection of subjects and methodology part. It is too brief and doesn’t provide quality justification of the purpose statement.

Validity issue – your study is not successful to provide validity evidence of your measurement issues. I am not really convinced to your findings and not clear about what would be potential lessons from reading your study.

Please check and re-confirm and have other experienced scholars to read your manuscript prior to “submission” in terms of “research process” and conclusion part. Your current conclusion is still too brief and not really meaningful.

Author Response

Dear reviewer,

We appreciate the time and effort that you have dedicated to providing your valuable feedback on our manuscript. We are grateful for your insightful comments on our paper. We have incorporated changes to reflect the suggestions provided. (The page and line numbers are addressed according to the clean version of the manuscript). All changes are marked in yellow in the revised manuscript. We hope will reach your aspirations. Thank you very much.

Here is a point-by-point response to the reviewers’ comments and concerns:

Reviewer 3

Glad to have an opportunity to review this manuscript, but first of all, this manuscript cannot be accepted as its current form and format. There are severe problematic areas of the manuscript and the authors were not able to deal with the essential aspects of so-called “scientific research.”

This study cannot be progressed into any further steps of publication in this quality journal unless the following issues properly dealt with:

1. The abstract not successfully compiling and summarizing focal points of this study

RESPONSE: Thank you very much for this comment. However, since this is already mentioned in the abstract, with the prevalence of diagnosed and undiagnosed T2D, in addition to the uncontrolled glycaemic rate for diagnosed T2D patients. Moreover, among those without a clinical diagnosis of T2D, we pointed out the strong association between daily added sugar intake and fasting glucose, and also the strong associations between added sugar intake and the risk of undiagnosed T2D throughout the stratified analysis.

2. The authors were not successfully provide adequate rationales for their comments in many of sentences especially when they commented about the selection of subjects and methodology part. It is too brief and doesn’t provide quality justification of the purpose statement.

RESPONSE: Thank you very much for this valuable comment. We have improved the methodology based on your suggestion, we have added more details on the subject selection, that the participants of the research were chosen from various clinics within the PHCs. We invited all the visitors during the working hours of those clinics from the early morning until afternoon ‘‘the participants of the research were chosen from various clinics within the PHCs. We invited all the visitors during the working hours of those clinics from the early morning until afternoon’’. (Page 2; line 85-87). Furthermore, we have added a paragraph with details for data collection and referred to the pilot study, which was conducted before the formal sample collection (page 3; line 104-112). ‘‘The initial version of the questionnaire was designed and then validated in two steps. First, the research instrument was sent to medical experts to express its simplicity, relativity, and importance. Secondly, a pilot study was conducted among a small group of PHC’s visitors (N=30) to verify the validity and reliability of the data collection process before the formal data collection. Minor changes were made to the data collection approach for some items by considering the results of the pilot analysis. Furthermore, medical experts validated and reviewed the updated questionnaire to reconfirm the acceptability of the methods used to achieve the research goals. In the formal data collection process, the final version of the questionnaire has been used, and the data from the pilot study were not used in the final analysis’’.

3. Validity issue – your study is not successful to provide validity evidence of your measurement issues. I am not really convinced to your findings and not clear about what would be potential lessons from reading your study.

RESPONSE: Thank you very much for this comment, we verified the validity of our measurements, throughout the pilot study, and it has approved by medical experts before the formal data collection. We mentioned that in the methods, (page 3; line 104-112) ‘‘the initial version of the questionnaire was designed and then validated in two steps. First, the research instrument was sent to medical experts to express its simplicity, relativity, and importance. Secondly, a pilot study was conducted among a small group of PHC’s visitors (N=30) to verify the validity and reliability of the data collection process before the formal data collection’’. On other hand, in our knowledge, our findings have a potential lesson, which they support a greater focus towards reducing the diabetes burden among Gaza residents, with the presence of the high prevalence of T2D and poor glycaemic control rate for the diagnosed T2D patients. Moreover, the exceeded consumption of daily added sugar among the Palestinians, with the strong association between daily added sugar intake and the risk of undiagnosed T2D throughout the stratified analysis for sex, age, BMI, and population.

4. Please check and re-confirm and have other experienced scholars to read your manuscript prior to “submission” in terms of “research process” and conclusion part. Your current conclusion is still too brief and not really meaningful.

RESPONSE: Thank you very much for providing this comment, we have re-confirmed our manuscript with other experienced scholars, particularly for the research process and conclusion parts. Additionally, the conclusion has been edited to be not briefed, we hope to be more informative and meaningful in the current version.

Round 2

Reviewer 3 Report

I am grateful for the possibility to revise this research study.

The prevalence of type 2 diabetes and its association with sugar intake in citizens and refugees in Gaza Strip, Palestine is a trend topic in the current research literature and may be a main focus of interest for readers.

This is a well-written manuscript with an important clinical message, and should be of great interest to the readers of International Journal of Environmental Research and Public Health. However, from my point of view, authors should include the following requeriments

Introduction may be improved adding new information in order to provide an adequate state-of-the-art including some references. I suggest to include this references include in the attached to complete this requirement related to Diabetic foot complications that authors do not included

Introduction section is deep enough with and adequate focus that may help readers to improve knowledge about the topic. However authors should improve the stay of art, for example including references to quality of life I suggest to include this references include in the atteched to complet this requeriment relative to frailty

- Navarro-Flores E, Romero-Morales C, Becerro de Bengoa-Vallejo R, et al. Sex Differences in Frail Older Adults with Foot Pain in a Spanish Population: An Observational Study. Int J Environ Res Public Health. 2020;17(17):. doi:10.3390/ijerph17176141

Or in the case or complications related to diabetic foot

-Navarro-Flores, E.; Morales-Asencio, J.M.; Cervera-Marín, J.A.; Labajos-Manzanares, M.T.; Gijon-Nogueron, G. Development, validation and psychometric analysis of the diabetic foot self-care questionnaire of the University of Malaga, Spain (DFSQ-UMA). J. Tissue Viability 2015, 24, 24–34.

Methods are well-designed with relevant and complete information. Correct sample size calculations, good description of the properties of the outcome measurements as well as detailed statistical analyses were included.

Discussion section is well structured with different sections. Authors manage well the discussion leading a good comparison with the showed references.

However, author should discuss the possible influence of aging a health releated quality of life on diabetes mellitus, as a consequence of diet style in this kind of patient,and their influence in psycosocial aspects, as the case of self care

Author Response

Nov 11th, 2020

Dear Professor Dr. Paul B. Tchounwou,

Thank you for allowing us to submit a revised draft of our Manuscript ID: ijerph-982125-Originally titled [Prevalence of Type 2 Diabetes and Its Association with Added Sugar Intake in Citizens and Refugees Aged 40 or Older in the Gaza Strip, Palestine]. We are grateful to the reviewers for their insightful comments on our paper. We have incorporated changes to reflect the suggestions provided by the reviewer/s. 

All changes for the second-round revision are marked in yellow. We hope this version would be acceptable for publication. Thank you very much.

Here is a point-by-point response to the reviewers’ comments and concerns:

Reviewer 3

(Round 2)

I am grateful for the possibility to revise this research study. The prevalence of type 2 diabetes and its association with sugar intake in citizens and refugees in Gaza Strip, Palestine is a trend topic in the current research literature and may be a main focus of interest for readers. This is a well-written manuscript with an important clinical message, and should be of great interest to the readers of International Journal of Environmental Research and Public Health. However, from my point of view, authors should include the following requirements.

RESPONSE: Thank you for the comments!

Introduction may be improved adding new information in order to provide an adequate state-of-the-art including some references. I suggest to include this references include in the attached to complete this requirement related to Diabetic foot complications that authors do not included

Introduction section is deep enough with and adequate focus that may help readers to improve knowledge about the topic. However authors should improve the stay of art, for example including references to quality of life I suggest to include this references include in the attached to complete this requirement relative to frailty

- Navarro-Flores E, Romero-Morales C, Becerro de Bengoa-Vallejo R, et al. Sex Differences in Frail Older Adults with Foot Pain in a Spanish Population: An Observational Study. Int J Environ Res Public Health. 2020;17(17):. doi:10.3390/ijerph17176141

Or in the case or complications related to diabetic foot

-Navarro-Flores, E.; Morales-Asencio, J.M.; Cervera-Marín, J.A.; Labajos-Manzanares, M.T.; Gijon-Nogueron, G. Development, validation and psychometric analysis of the diabetic foot self-care questionnaire of the University of Malaga, Spain (DFSQ-UMA). J. Tissue Viability 2015, 24, 24–34.

RESPONSE: Thanks for your suggestion! We have added those references accordingly at (Page 2; line 67-71( “On the other hand, diabetic patients with high added sugar intake can develop diabetic foot ulcers in the long term, especially for elderly adults [24]. It may have a negative impact on the quality of life and diabetes self-management [25]. Thus, the diabetic patient should have special management of his nutrition, including added sugar, which has a critical role in the wound healing process [26].”

Methods are well-designed with relevant and complete information. Correct sample size calculations, good description of the properties of the outcome measurements as well as detailed statistical analyses were included. Discussion section is well structured with different sections. Authors manage well the discussion leading a good comparison with the showed references.

RESPONSE: Thank you for the comments!

However, author should discuss the possible influence of aging a health related quality of life on diabetes mellitus, as a consequence of diet style in this kind of patient, and their influence in psychosocial aspects, as the case of self-care.

RESPONSE: Thank you for your suggestion. We agree and have added a relevant paragraph in this regard at (Page 12; lines 263-268( “On the other hand, it is already mentioned in the literature that aging influences the health-related quality of life for diabetic patients [46]. A survey also suggested that T2D patients aged 65 to 84 years, had a poorer health-related quality of life (HRQOL) than the general population, especially regarding different aspects of physical functioning and emotional disturbances [47]. Additionally, older adults with diabetes have unique issues that may lead to deterioration in their self-care behaviors [48].”

Dear reviewers, thank you again for your constructive and valuable comments, directions, and suggestions during the first round, they made a great improvement to the paper. We highly appreciate that. We look forward to hearing from you in due time regarding our submission and to respond to any further questions and comments you may have.

Sincerely,